# Molecular Epidemiology and Virulence of Non-Typhoidal *Salmonella* in Armenia

**DOI:** 10.3390/ijms23169330

**Published:** 2022-08-18

**Authors:** Anahit Sedrakyan, Zhanna Ktsoyan, Karine Arakelova, Zaruhi Gevorgyan, Magdalina Zakharyan, Shoghik Hakobyan, Alvard Hovhannisyan, Arsen Arakelyan, Rustam Aminov

**Affiliations:** 1Institute of Molecular Biology, National Academy of Sciences of RA, Yerevan 0014, Armenia; 2Department of Clinical Laboratory Diagnostics, Yerevan State Medical University after M. Heratsi, Yerevan 0025, Armenia; 3School of Medicine, Medical Sciences and Nutrition, University of Aberdeen, Aberdeen AB25 2ZD, UK

**Keywords:** non-typhoidal *Salmonella*, ERIC-PCR typing, WGS, virulence-related genes, SPIs, *Salmonella* virulence plasmids, prophages

## Abstract

In this work, we analysed human isolates of nontyphoidal *Salmonella enterica* subsp. *enterica* (NTS), which were collected from salmonellosis cases in Armenia from 1996 to 2019. This disease became a leading food-borne bacterial infection in the region, with the younger age groups especially affected. The isolates were characterised by serotyping, Enterobacterial Repetitive Intergenic Consensus (ERIC-PCR) typing, and whole genome sequencing (WGS). The main serotypes were *S*. Typhimurium, *S*. Enteritidis, and *S*. Arizonae. ERIC-PCR indicated a high degree of clonality among *S*. Typhimurium strains, which were also multidrug-resistant and produced extended spectrum beta-lactamases. During the study period, the frequency of *S*. Typhimurium and *S*. Arizonae isolations decreased, but with the increase in *S*. Enteritidis and other NTS. A total of 42 NTS isolates were subjected to WGS and explored for virulence-related traits and the corresponding genetic elements. Some virulence and genetic factors were shared by all NTS serotypes, while the main differences were attributed to the serotype-specific diversity of virulence genes, SPIs, virulence plasmids, and phages. The results indicated the variability and dynamics in the epidemiology of salmonellosis and a high virulence potential of human NTS isolates circulating in the region.

## 1. Introduction

The current classification of the genus *Salmonella* includes two species, *S. bongori* and *S. enterica*, with the latter consisting of the following six subspecies: *arizonae*, *diarizonae*, *enterica*, *houtenae*, *indica*, and *salamae* [1]. Recently, based on genomic data, it has been proposed to extend the number of subspecies within *S. enterica* to include subspecies *londinensis*, *brasiliensis*, *hibernicus*, *essexiensis*, and *reptilium*, while elevating *S. enterica* subspecies *arizonae* to the species level, *S. arizonae* [2]. The majority of *Salmonella* infections in warm-blooded hosts are caused by various serotypes of *S. enterica* subsp. *enterica*, while other subspecies and *S. bongori* are largely isolated from the environment or cold-blooded organisms [3]. The typhoidal (Typhi) and paratyphoidal (Paratyphi A, B and C, and Sendai) serotypes of *S. enterica* subsp. *enterica* are human-adapted and cause invasive extra-intestinal disease. Other serotypes within this subspecies are known as nontyphoidal *Salmonella* (NTS), which, in contrast, are characterised by a broader host range and usually cause self-limiting gastroenteritis. NTS are the most common causes of human food-borne outbreaks and diseases worldwide, with an estimated 78.7 million cases per year [4]. In the USA, NTS is estimated to cause 1.2 million food-borne illnesses each year, with 23,128 hospitalisations and 452 deaths [5].

*S*. Typhimurium and *S*. Enteritidis are the two serotypes within *S. enterica* subsp. *enterica* that are the most widespread, transmitted from animals to humans through contaminated food products, and responsible for the majority of clinical cases of salmonellosis in most parts of the world [6,7,8]. There are regional differences in serotype frequencies though, for example, with the prevalence of *S*. Typhimurium in Australia and Argentina and *S*. Enteritidis in Brazil, Tunisia and Europe, while in a number of countries these two serotypes are represented more equally [3].

In many food-production animals, the carriage of NTS usually does not manifest any substantive clinical presentation, and thus these carriers can be considered as NTS reservoirs, from which this infectious agent could be disseminated into a wider environment, including contamination of human food commodities [9]. However, uncovering the epidemiological link between NTS strains in food-producing animals and human disease is not always straightforward. These strains must be successful colonisers of the gastrointestinal or reproductive tract of the reservoir hosts, retain viability in the environment and in the food chain outside of the host organism, survive the extremes of food preparation and host defences to reach the human small intestine, and finally break the colonisation resistance and immune barriers to invade and multiply in the host cells to cause the disease. While we have fairly good knowledge and mechanistic explanations for the human disease part, the behaviour of these pathogens in other ecological compartments remains poorly understood. With the advent of genomic technologies, however, it is now possible to identify genomic signatures of NTS that are associated with host-range, tissue tropism or differential virulence [10]. The mechanisms of NTS survival in the environment and the food chain, however, still have to be addressed in greater detail.

The clinical picture and outcome of salmonellosis is characterised by a complex host–microbe interaction. The efficiency of the host response to NTS infection depends on many factors, including immune and nutritional status, age, gastric pH, genetic susceptibility and both the innate and adaptive arms of immunity [3]. On the bacterial side, the factors that contribute to disease severity are determined by serotype, infectious dose, injury of bacterial cells, antimicrobial resistance (AMR), gene inactivation, and virulence factors.

The presently known virulence factors in *Salmonella* include the production of toxins, capsules, flagella, fimbriae, secretion systems for the export of effector proteins and other factors, which are involved in various stages of infection. Genetically, these factors are encoded by *Salmonella* Pathogenicity Islands (SPIs), virulence plasmids and (pro)phages, or are chromosomally located. Currently, there are 24 SPIs identified in *Salmonella*, which are involved in the different stages of infection [3]. Genetic and phenotypic attributes are most thoroughly studied in two of them, SPI-1 and SPI-2. SPI-1 is widely present in both *Salmonella* species, *S. enterica* and *S. bongori*, and in many subspecies. It encodes a type three secretion system (T3SS), which provides the translocation of effector molecules involved in the invasion of host cells. SPI-2, which is present in many subspecies of *S. enterica* but not in *S. bongori*, encodes an additional T3SS that is involved in the translocation of effector molecules important for the intracellular survival of these bacteria. Other SPIs are variably present in *S. enterica* subspecies, with some encoding other secretion systems such as T1SS and T6SS, other effector molecules, and fimbriae.

Virulence plasmids present only in a limited number of NTS serotypes, including *S.* Typhimurium (pSTV/pSLT, 94.7 Kb) and *S.* Enteritidis (pSEV, 60 Kb) [11]. The low copy-number plasmids pSTV/pSLT and pSEV are non-conjugative but can be mobilised by F plasmids. The enhanced virulence phenotype conferred by these plasmids is due to the presence of the *spvRABCD* operon, which encodes proteins that destabilise the host cytoskeleton structure thus increasing bacterial survival. Besides, these plasmids also carry additional virulence genes such as *rck* (resistance to complement killing), *pef* (plasmid-encoded fimbriae), *srgA* (SdiA-regulated gene, putative disulphide bond oxidoreductase), and *mig-5* (macrophage-inducible gene coding for putative carbonic anhydrase) [12]. Additionally, a large self-transmissible plasmid pUO-StVR2 (140 Kb) was isolated from clinical strains of *S.* Typhimurium, which encodes *spv* and *rck* virulence genes and resistance towards quaternary ammonium compounds, mercury, and antimicrobials (ampicillin, streptomycin-spectinomycin, sulfadiazine, chloramphenicol, and tetracycline) [13].

Flagella assist the movement towards the epithelial cells of the host, and they are also potent inducers of the host immune response [14]. *Salmonella* serovars use the induction of host inflammatory responses to evade competition and create a novel nutrient niche with inflammation-derived nutrients to ensure propagation and dissemination [15]. Fimbriae are synthesised by many Gram-negative and Gram-positive bacteria, and they serve as important virulence factors in NTS aiding in attachment and adhesion to the host cells [3]. A recent analysis of 242 genomes of *S. enterica* subsp. *enterica* human and animal isolates, belonging to 217 serotypes, detected 2894 chaperone-usher (CU)-type fimbrial usher sequences [16]. Thus, an average isolate possesses 12 different CU fimbrial ushers, in the range of 6 to 18 per genome. This study suggested that most CU fimbriae are broadly distributed and their acquisition occurs before the divergence of *S. enterica* subsp. *enterica*. Diversity of CU fimbriae within the subspecies appeared to be due to differences in the phylogenetic clade rather than the reflection of host-driven selection. This analysis also indicated that plasmids are primarily responsible for the horizontal exchange and the observed diversity of CU fimbriae in these bacteria.

As noted before, gastroenteritis cases due to NTS worldwide are predominantly caused by two serotypes, *S.* Typhimurium and *S.* Enteritidis, and this is also the case in Armenia. We reported previously that *S*. Typhimurium was the most frequently encountered NTS serotype among gastroenteritis patients in Armenia in 1996–2006, accounting for up to 55.7% of all NTS-caused diseases [17,18,19]. There was also a tendency for an increasing number of diseases caused by *S*. Enteritidis, with the isolates characterised by a higher susceptibility to antimicrobials (AMs) and with a less frequently encountered multidrug-resistant (MDR) phenotype compared to *S*. Typhimurium isolates [20,21]. We also reported serotype-specific features of the disease, susceptibility to AMs and prevalence of MDR phenotypes among the human NTS isolates circulating in Armenia [17,18,19,22].

The aims of the present study were to extend the range of the epidemiological analyses up to the year 2019 and analyse our collection of NTS isolates further with the use of Enterobacterial Repetitive Intergenic Consensus (ERIC-PCR) and whole genome sequencing (WGS). WGS data were further used to uncover the genetic background of virulence in our NTS isolates, the association of virulence genes with mobile genetic elements (MGEs), and to obtain serotype information for the isolates that resisted conventional agglutination techniques of serotyping. Thus, the present work is the first molecular epidemiological study of human NTS isolates in this geographic region. It contributes new data to the epidemiology and dynamic of this important food-borne pathogen within the international context.

## 2. Results

### 2.1. Incidence of Salmonellosis in Armenia in 2003–2021

According to the Statistical Committee of the Republic of Armenia (SCRA), salmonellosis was second to shigellosis as the most common bacterial gastroenteritis, which required hospitalisation, in Armenia from 2003 to 2018 [23]. However, since 2019, salmonellosis has replaced shigellosis as the leading food-borne bacterial infection in Armenia. The available data on gastroenteritis caused by NTS in Armenia for the period from 2003 to 2021 (Figure 1) indicate the variability in the total number of confirmed cases per year, with a median of 384 (interquartile range (IQR) 293–451, range 199–793). According to the SCRA data, the incidence rate of salmonellosis per 100,000 persons between 2003 and 2021 demonstrated a substantial increase from 6.2 in 2003 to 26.7 in 2019 [23].

It should be emphasised here that among the patients with salmonellosis there was a high proportion of the younger age groups, under 14 and 18 years old (Figure 1). In 2003–2015, the estimated incidence rate per 100,000 was significantly higher in the age group under 14 compared with the remaining age groups combined, of 35.7 (IQR 27.55–41.2, range 18.8–47.8) and 5.77 (IQR 3.67–6.55, range 2.131–8.646), correspondingly (Mann–Whitney test, *p* < 0.0001). In 2016–2021, the incidence rate in the age group under 18 was also higher as compared with all other age groups combined: 42.3 (IQR 35.3–57.1, range 33.8–73.9) and 6.29 (IQR 4.87–9.72, range 5.01–11.26), respectively (Mann–Whitney test, *p* = 0.002).

### 2.2. Serotypes of NTS Circulating in Armenia in 2011–2017

In this study, the serotyping of 1423 NTS isolates, recovered from patients admitted to the Nork Infectious Clinical Hospital (NICH) (Ministry of Health, Armenia) between 2011 and 2017, was performed (Figure 2). It should be emphasised here that the number of NTS isolates in this analysis accounted for the majority of confirmed salmonellosis cases in Armenia, with 53.16% of all cases during the period from 2011 to 2017 analysed (1423 isolates from 2677 cases).

The results indicated that the most common NTS serotype in 2011–2017 was *S*. Enteritidis (26.99%, 384 cases) and the second most common serotype was *S*. Typhimurium (14.97%, 213 cases), followed by *S*. Arizonae (7.31%, 104 cases). These three serotypes accounted for nearly half of the NTS infections during the study period (49.26%, 701 out of 1423 cases). The proportion of infections caused by other NTS serotypes ranged from 40.6% to 44.6% in 2011–2014, with a substantial increase in 2015–2017, accounting for more than 50% of NTS isolates (56.1–72.1%). Remarkably, the ratio of infections caused by *S*. Enteritidis vs. *S*. Typhimurium increased more than five-fold during the seven-year observation period, from 1.12 in 2011 to 6.03 in 2017. Thus, the proportion of *S*. Typhimurium and *S*. Arizonae isolates gradually decreased, while *S*. Enteritidis and other NTS serotypes became leading causes of the disease in the region.

### 2.3. ERIC-PCR Subtyping of Human NTS Isolates Circulating in Armenia

ERIC-PCR was used to subtype 112 human isolates of NTS. These isolates were selected as representatives from our collection of 316 NTS isolates recovered from the gastroenteritis patients admitted to the NICH between 1996 and 2019. Selection criteria for isolates in this analysis included the serotype, year of isolation, and antimicrobial resistance (AMR) profile. Among the selected isolates, *S*. Typhimurium isolates were the most represented (48 MDR isolates and 12 non-MDR isolates), given the high prevalence of MDR among the isolates of this serotype in our collection of human NTS from Armenia [19,20,22]. One of these isolates, *S*. Typhimurium A_684, was isolated from the blood culture of a 4-month-old infant in 1996. The second group consisted of 31 *S*. Enteritidis isolates, including all four isolates displaying the MDR phenotype. In addition, 21 isolates (4 MDR and 17 non-MDR) with no serotype information were included in the third group for ERIC-PCR typing. The ERIC-PCR analysis of 112 NTS isolates yielded differential patterns consisting of 5–19 bands (Figure 3).

The dendrogram generated from ERIC-PCR typing demonstrated that *S*. Typhimurium strains were mostly grouped together, showing at least an 80% similarity of the band patterns. Among the 60 *S*. Typhimurium strains analysed, 55 of them were grouped into eight clusters (I and IV–X), suggesting the clonal structure of isolates within these clusters (Figure 3). The largest cluster, designated IV, included 22 *S*. Typhimurium strains collected between 1996 and 2016, suggesting the circulation of clones that were reproductively successful and therefore could be re-isolated within the period of two decades. With the exception of two *S*. Typhimurium strains, which did not display an MDR phenotype, the others were MDR, with 18 MDR strains showing a similar or identical AMR resistance profile. Interestingly, five isolates with no serotype information were also grouped within this cluster with 100% similarity, suggesting that these strains belong to *S*. Typhimurium serotype. They were drastically different in terms of AMR phenotypes though as only a single strain was intermediately resistant to nalidixic acid, while all others were sensitive to all AMs tested. Notably, another cluster V, which also included MDR *S*. Typhimurium strains, showed 97.91% similarity with the largest cluster IV. It should be emphasised here that at least 60% of *S*. Typhimurium isolates, which were grouped in clusters IV and V, were ESBL-producers. These findings suggest that MDR *S*. Typhimurium strains showing at least 97% similarity under ERIC-PCR typing have been circulating in Armenia for 20 years, from 1996 to 2016. Two other large clusters with identical ERIC-PCR profiles within *S*. Typhimurium, clusters I (9 isolates) and IX (8 isolates), also unified strains with MDR phenotypes. Other smaller MDR clusters included VIII and X. Of note, the bloodstream *S*. Typhimurium A_684 isolate, which also exhibited the MDR phenotype, showed 95.8% similarity to cluster IX.

The strains of *S*. Enteritidis in this study demonstrated more variable ERIC-PCR profiles and thus were grouped together with a lesser degree of similarity (Figure 3). Among the thirty one isolates belonging to the serotype Enteritidis, twenty isolates were grouped into seven clusters showing identical ERIC profiles (XI–XV, XVII–XIX). The largest cluster of *S*. Enteritidis (XIV) includes six isolates displaying a non-MDR phenotype. In general, AMR and MDR phenotypes among *S*. Enteritidis strains were much less prominent compared to *S*. Typhimurium strains. Interestingly, the four MDR *S*. Enteritidis isolates in this study were not grouped together in a cluster of clonal isolates, but were dispersed within different clusters.

The NTS strains with no serotype information initially consisted of 21 isolates. On the basis of WGS (see below), five of them were identified as *S.* Newport, *S.* Agona, *S.* Derby (two isolates), and *S.* Kentucky, while the remaining sixteen were designated as “NI” in Figure 3. These strains were basically scattered across the tree. From the 16 “NI” strains, 11 of them grouped well within the corresponding *S*. Typhimurium clusters and could be potentially considered as belonging to this serotype. One strain, 7243, displayed an ERIC-PCR profile identical to the *S*. Enteritidis strain A_6059 and was assembled within cluster XVII. The remaining four strains are potentially represented by serotypes different from those discussed above. Since one of the selection criteria for further analysis was AMR/MDR phenotype, all isolates belonging to the serotypes Newport, Derby, and Kentucky displayed the MDR phenotype (Figure 3).

For comparative purposes, MLST information, which was generated from WGS data (see below), was incorporated into Figure 3. According to the MLST typing, all *S*. Typhimurium isolates were classified into two sequence types (STs), ST328 and ST19, and *S*. Enteritidis—into a single sequence type, ST11. The use of ERIC-PCR typing, therefore, offered additional discriminatory power and provided more detailed insights into the epidemiology of NTS strains. The results of ERIC-PCR typing indicated a high degree of clonality among the human *S*. Typhimurium isolates that have been circulating in the region for two decades, while *S*. Enteritidis strains were characterised by later dates of isolation and displayed a relatively higher degree of genetic variability.

### 2.4. Whole Genome Sequencing of Human NTS Isolates

A total of 42 NTS isolates from Armenia were subjected to WGS. For this work, the isolates belonging to the most common serotypes *S*. Enteritidis (7 strains) and *S*. Typhimurium (30 strains) were selected based on AMR/MDR profile and year of isolation to cover the study period evenly. ERIC-PCR results were used to avoid the redundant sequencing of clonal isolates. In addition, isolates with no serotype information were also analysed by WGS. The general information concerning the genomes and genomic characteristics of NTS isolates was acquired using Genomics tools of the Bacterial and Viral Bioinformatics Resource Center (BV-BRC) (https://www.bv-brc.org, accessed on 15 May 2022) and presented in Appendix A. All genome sequences are available through the ENA database under the project accession number PRJEB36290. Individual accession numbers are listed in Appendix A. The general genomic features such as genome sizes of NTS isolates were in the range of 4.7–5.1 Mb, with GC-content in the range of 51.92–52.19%, which is consistent with the genomic features of the reference strain, *S. enterica* subsp. *enterica* serovar Typhimurium LT2 [24].

### 2.5. In Silico Serotyping and MLST Analysis of NTS Isolates

The serotypes of the 42 sequenced isolates from Armenia were identified from WGS data using the SeqSero2 ([25,26]; www.denglab.info/SeqSero2, accessed on 22 July 2022) web-based tool. The serotypes of all sequenced *S*. Typhimurium and *S*. Enteritidis isolates were confirmed by WGS analysis. Interestingly, we did not encounter any monophasic *S*. Typhimurium isolates in our study (Appendix A). On the basis of WGS data, five NTS isolates, the serotype of which was not possible to determine using the conventional agglutination tests, were typed as Derby (two MDR isolates), Agona, Kentucky (MDR isolate), and Newport (MDR isolate) (Figure 3).

Additionally, on the basis of WGS data, the STs of the isolates were determined in silico using the PubMLST website and databases [27]. The WGS-based serotype and ST information of 42 NTS isolates is shown in Figure 4.

The sequenced *S*. Typhimurium isolates belonged either to ST328 (22 MDR and 4 non-MDR strains) or ST19 (2 MDR and 2 non-MDR strain). Interestingly, all *S*. Typhimurium isolates within the ST19 lineage were isolated exclusively in 2016, while no ST19 representative could be identified among the NTS strains collected before 2016. All *S*. Enteritidis isolates in this study were assigned to ST11, regardless of their AMR/MDR phenotype or year of isolation. The isolates belonging to other NTS serotypes were classified as follows: two MDR *S*. Derby isolates as belonging to ST40; one *S*. Agona isolate—to ST13; one MDR *S*. Kentucky isolate—to ST198; and one MDR *S*. Newport isolate—to ST31.

The results emphasised the importance of the WGS-based approach for the identification of epidemiologically relevant NTS serovars, including strains that cannot be serotyped using conventional agglutination tests. 

### 2.6. Virulence-Related Traits and Their Genetic Constituents in NTS Isolates

#### 2.6.1. Virulence Genes

Genomes of 42 NTS isolates were interrogated for the presence of known and putative virulence-related genes using the Virulence Factor Database (VFDB) [28] and Pathosystems Resource Integration Center (PATRIC) [29] web resources. A summary of the results is presented in Table 1, Appendix A. The total number of the virulence-related genes identified by VFDB and PATRIC was in the range of 155–177 and 309–344, respectively (Appendix A).

According to the VFanalyzer [28] results, the following genes encoding fimbrial adherence determinants were found in all our NTS isolates, irrespective of the serotype: *csg*, *bcf*, *fim*, *stb*, *std*, *stf*, *sth*, and *sti*; whereas the presence of some others was serotype-specific (Table 1 and Appendix A). In particular, the *peg* operon was detected in *S*. Enteritidis, *S*. Derby, and *S*. Newport isolates, while it was absent in all other serotypes. The *sef* operon was found only in *S*. Enteritidis isolates, while the *sta* genes were found in *S*. Derby and *S*. Agona isolates. Notably, *stk* and *tcf*, as well as *faeCDEHIJ* (encoding K88 fimbriae in *Escherichia coli*) were detected only in an *S*. Kentucky isolate. The complete *stcABCD* operon was detected only in *S*. Typhimurium and *S*. Agona isolates, and it was truncated to *stcBC* genes in *S*. Kentucky. On the other hand, the *ste* operon was absent in *S*. Typhimurium isolates, *lpf* was absent in *S*. Derby, *stj* was absent in *S*. Enteritidis and *S*. Derby, and the *safA* gene was absent in *S*. Agona. Among the genes encoding for nonfimbrial adherence determinants, *misL*, *shdA*, and *sinH* were shared by all isolates, while *ratB* was absent in *S*. Derby and *S*. Kentucky isolates. In addition, the prevalence of *pef* operon was both serotype- and isolate-specific, as the operon was found in 85.7% of *S*. Enteritidis and 13.3% of *S*. Typhimurium isolates, while it was absent in isolates of other serotypes.

The genes, which encode type III secretion systems (T3SS) and located within SPI1 and SPI2, were detectable in all our NTS isolates, irrespective of the serotype. These genes included *hil*, *iacP*, *iagB*, *inv*, *org*, *prg*, *sic*, *sipD*, *spa*, *slrP*, *sprB*, *ssa*, *sse*, and *ssr*. Additionally, all the NTS isolates shared the following set of effectors: T3SS-1 translocated *avrA*, *sipABC*, *sopA*, *sopB*/*sigD*, *sopD*, *sopE2*, and *sptP*; T3SS-2 translocated *pipB2*, *pipB*, *sifA*, *spiC*/*ssaB*, *sseFGL*, and *sspH2.* At the same time, the effector encoded by the *sopE* gene and translocated by T3SS-1 was detected only in *S*. Enteritidis and *S*. Newport isolates. The serotype-specific distribution was also identified for some of the T3SS-2-translocated effectors. Particularly, the *gogB* gene was found in *S*. Typhimurium isolates only, while the *sopD2* gene was absent in this serotype. The *sseI*/*srfH* gene was common in *S*. Typhimurium and *S*. Enteritidis isolates but was not detected in all other serotypes. Moreover, the *sseK1* gene was absent in the *S*. Newport isolate and the *sseK2* gene was absent in the *S*. Agona isolate. 

In addition, macrophage inducible *mig14*, magnesium uptake encoding *mgtBC* and genes of the two-component regulatory system *phoPQ* were detected in all our NTS isolates. The oxidative stress adaptation virulence factor *sodC1* was found in *S*. Typhimurium and *S*. Enteritidis, while it was absent in other serotypes. Other virulence genes such as *spvBCD*, *pefABCD*, *mig-5* and *rck* were detected mainly in *S*. Enteritidis (85.7% of isolates) and *S*. Typhimurium (13.3% of isolates), while absent in other serotypes.

The latter observation is in agreement with the results of annotation of our NTS isolates using the web resource PATRIC [29], which confirmed the presence of the *spvABCDR*, *pefCD* and *rcK* genes in the same isolates (Appendix A). The serotype-specific distribution of the *lpf*, *sopE*, *sseI* and *stcCD* genes was detected by both VFanalyzer and PATRIC.

Special attention was paid to the composition of virulence-related genes in the *S*. Typhimurium A_684 isolate that was isolated from blood culture. Interestingly, the following six genes were absent in this isolate: *sifB* encoding T3SS-2 translocated effector, *steA* encoding secreted effector protein, *yncB* encoding putative oxidoreductase, *soxS* encoding DNA-binding transcriptional dual regulator, *pdgL* encoding D-alanyl-D-alanine dipeptidase (EC 3.4.13.22), and *eptA* encoding Lipid A phosphoethanolamine transferase (EC 2.7.8.43). Correspondingly, this isolate had the smallest genome size among *S*. Typhimurium isolates in this study (4,807,508 bp; Appendix A), which was about 170.5 kb smaller compared to the average genome size of the other ST328 isolates (4,978,018 bp).

#### 2.6.2. *Salmonella* Pathogenicity Islands

*Salmonella* pathogenicity islands (SPIs) in the genomic sequences of our NTS isolates were identified using the SPIFinder 2.0 tool [30] and the results are shown in Appendix A. A total of 14 SPIs were detected according to this analysis. The SPI-1 to SPI-5 and SPI-9 were shared by all NTS isolates in this study, regardless of the serotype. One SPI designated as “Not named” by SPIFinder, which contained a putative pathogenicity island with the *ssaD* gene (GenBank accession number JQ071613), was also present in all isolates. The prevalence of all other SPIs detected was serotype-specific. The SPI-10 (*S*. Gallinarum strain SGE-3 pathogenicity island carrying the fimbrin-like protein *sefD* gene (GenBank accession number AY956839) was found in *S*. Enteritidis isolates only. According to the results of annotation by SPIFinder, C63PI (GenBank accession number AF128999) was absent in *S*. Typhimurium isolates, while it was detected in all other NTS isolates.

More profound variations in SPI profiles were identified in *S*. Derby, *S*. Agona, and *S*. Kentucky as compared to other serotypes. In particular, the SPI-13, SPI-14 and CS54 islands were detected in all our NTS isolates, with the exception of *S*. Derby, *S*. Agona, and *S*. Kentucky. Moreover, SPI-8 was found in the *S*. Agona isolate only. Notably, all *S*. Enteritidis isolates had an identical profile to the SPIs detected. The SPI profile of two *S*. Derby isolates was also identical. On the contrary, *S*. Typhimurium isolates were more heterogenous in terms of the SPIs detected, which were ST-dependent. In particular, *S*. Typhimurium ST19 isolates carried SPI-1 variants (GenBank accession numbers AF148689 and U16303), which were absent in all *S*. Typhimurium ST328 isolates. 

Remarkably, the high-pathogenicity island (HPI) (GenBank accession numbers FJ212115 and FJ212116) was identified in one isolate, *S*. Typhimurium A_7417 (ST328, non-MDR, isolated in 2011). This isolate was chosen for WGS because of the unusual immune response in the patient infected by it whereby the systemic level of IL-17 was three-fold and ten-fold lower than in healthy and *S*. Typhimurium-infected patients, respectively. This HPI, which encodes a yersiniabactin-mediated iron acquisition system, was initially described in highly pathogenic strains of *Yersinia* and then in several members of the Enterobacteriaceae [31]. The contig, in which this HPI was located, was analysed using VFanalyzer and BLAST and the following virulence-related genes were identified: *irp1*, *irp2*, *fyuA*, and *ybtAEPQSTUX*. The region carrying these genes had a high homology to the region of 27.8 kb in *E. coli* strain Es_ST410_NW1_NDM_09_2017 (GenBank accession number CP031231.1), with an Average Nucleotide Identity (ANI, [32]) of 99.99%. These findings suggested the past horizontal transfer of virulence genes to NTS strains, which therefore acquired additional pathogenic properties.

#### 2.6.3. *Salmonella* Virulence Plasmids

All genomes were explored for the presence of virulence-associated plasmids using the PlasmidFinder 2.1 tool [33]. Results are presented in Appendix A. We detected *Salmonella* virulence plasmid-specific sequences in six out of seven *S*. Enteritidis genomes, but with a much lower prevalence of these sequences in *S*. Typhimurium genomes. They were detected only in four out of the thirty genomes sequenced. No virulence-associated plasmid sequences were detected in the genomes of other NTS serotypes (Agona, Derby, Kentucky, and Newport). It should be noted here that all *S*. Typhimurium isolates harboring virulence plasmids (2 MDR and 2 non-isolates) were isolated in 2016 and assigned to ST19. In genomic sequences of 26 *S*. Typhimurium ST328 isolates, the virulence-associated plasmids were not detectable irrespective of the year of isolation (1996–2016) or AMR/MDR phenotype.

The two *Salmonella* virulence replicons, which were identified within the same contigs, were IncFIB(S) (GenBank accession number FN432031) and IncFII(S) (GenBank accession number CP000858). All contigs that contained the plasmid-specific replicons predicted by PlasmidFinder 2.1 tool, were analysed by BLAST, VFanalyzer [28] and PATRIC [29]. All these contigs carried two replicons and encoded the virulence-associated *spvABCD*R and *pefABCD* operons, as well as *rck* and *mig-5* genes. The serotype-specific sizes of *Salmonella* virulence plasmids were identified in *S*. Typhimurium (~94 kb) and *S*. Enteritidis (56.5–59.3 kb) isolates (Appendix A). The virulence plasmids identified in *S*. Typhimurium isolates showed a high similarity to plasmid pSLT931 of *S*. Typhimurium strain 3-931 (GenBank: CP016390.1) and plasmid pSTV-Mu1 of *S*. Typhimurium belonging to ST19 (GenBank: KX777254.1), with the ANI value in the range of 99.88–99.94% (Appendix A). The virulence plasmids detected in our *S*. Enteritidis isolates demonstrated a high similarity to plasmid pSENV of *S*. Enteritidis strain A1636 (GenBank: CP063709.1) and plasmid pSEN of *S*. Enteritidis str. P125109 PT4 (GenBank: HG970000.1), with ANI values of 99.99–100%.

Other contigs in which the plasmid-specific sequences were predicted, were also analysed by VFanalyzer and these contigs were all negative for virulence-associated genes (data not shown). Thus, our results indicated the high prevalence of *Salmonella* virulence plasmids in human isolates of *S*. Enteritidis and in *S*. Typhimurium isolates belonging to ST19.

#### 2.6.4. Prophages Carrying Virulence-Related Genes

The genomic sequences were interrogated for the presence of prophage regions encoding virulence-related genes using PHASTER (PHAge Search Tool Enhanced Release) [34,35] and VFanalyzer [28]. A total of 13 intact prophages showing a serotype-specific distribution were identified in the genomes of our NTS isolates (Appendix A). According to VFDB analyses, only four of these prophages were carrying virulence-related genes.

One of them, phage Gifsy_1 (NC_010392), was detected in 93.3% (28/30) of *S*. Typhimurium isolates as well as in an *S*. Newport isolate. In 63.3% (19/30) of *S*. Typhimurium isolates, this phage harboured the *gogB gene*, while in the remaining *S*. Typhimurium isolates the gene was identified in the regions containing phage residues detected by BLAST analysis.

Another phage carrying virulence-related gene, Gifsy_2 (NC_010393), was identified in all *S.* Typhimurium and *S.* Enteritidis isolates, while not found in other serotypes. The *sodC1* gene was located in the phage region in all these isolates, whereas the additional *grvA* gene was found in *S*. Typhimurium isolates only. Moreover, in 76.7% (23/30) of *S*. Typhimurium isolates the *sseI/srfH* gene was also located within the intact Gifsy_2 phage, while in the remaining *S*. Typhimurium isolates and in all *S*. Enteritidis isolates this gene was located within the incomplete (defective) phages.

The phage Salmon_118970_sal3 (NC_031940) was identified in all but one of the *S*. Typhimurium isolates, as well as in *S*. Enteritidis and *S*. Derby isolates. According to results of annotation using VFanalyzer, the csgDEFG and *sseK2* genes were located in this phage region in all *S*. Enteritidis isolates. Among the *S*. Typhimurium isolates in this study, the *sseK2* gene was found within the phage region in four isolates belonging to ST19, while it was not phage-located in ST328 isolates.

Finally, the intact phage Salmon_SEN8 (NC_047753) carrying the *iroB* and *iroN* genes was detected only in an *S*. Agona isolate.

The virulence-related genes were not detected in all other intact prophages, which were identified within the NTS genomes using the PHASTER tool. Some virulence-related genes, however, were identified as being located within the incomplete phage regions. In particular, the *gtrAB* genes were located within the incomplete phage regions in all serotypes except *S*. Newport. In *S*. Derby isolates, the *sopE2* gene was also identified within a defective phage.

Thus, several phages that carry virulence-related genes are widespread in NTS isolates in Armenia. However, some virulence-related genes are located within defective phages and unlikely to be transferable.

## 3. Discussion

Salmonellosis represents one of the most common cases of bacterial gastroenteritis in Armenia over the past two decades. Furthermore, since 2019 salmonellosis has become the leading food-borne bacterial infection in the region, with the age groups of under 14 and 18 years old comprising at least 50% of patients. At present, *S*. Enteritidis has become one of the most common NTS serotypes in the region, with a substantial decrease in the proportion of *S*. Typhimurium and *S.* Arizonae infections. In the USA, the most common contaminated food commodities, which are responsible for more than 80% of outbreaks and which are caused by serotypes Enteritidis, Heidelberg, and Hadar, were attributed to eggs or poultry [36]. It was observed that the targeted elimination of *S*. Gallinarum in chicken in the USA led to the parallel sudden increase in *S*. Enteritidis, which may persist in chicken without clinical signs of disease [37]. Thus, the successful elimination of one NTS serotype may lead to the occupation of the vacant ecological niche by another NTS serotype. The cause, however, for the gradual increase in *S*. Enteritidis infections in Armenia is not clear. We may only speculate that some new ecological niches appeared in the regional food production systems such as poultry, which were successfully occupied by *S*. Enteritidis to become reservoirs for its further transmission through contaminated food products.

Another contributing factor, on the global scale, could be the general increase in *S*. Enteritidis infections originating from poultry, which was initially detected in North America, South America and Europe in the 1980s, and in the 1990s—in Asia and Africa [38,39,40]. It was suggested that the main contributing factor to the global spread of *S*. Enteritidis is the current system of poultry production, with centralised sourcing and international trade of poultry breeding stocks [41]. These conclusions were based on the large-scale phylodynamic analysis of the global *S*. Enteritidis populations with over 30,000 genomes from 98 countries during 1949–2020 and international trade of live poultry from the 1980s to the late 2010. Similarly to other international poultry production systems, the local producers in Armenia adhere to industry standards and depend on the supply of breeding stocks for the productions farms. 

Recently, there has also been an increase in the infections caused by other NTS serotypes. This observation supports an urgent need to expand the range of NTS serotypes that can be identified by conventional agglutination tests, beyond the current panels limited to the most common serotypes. Similarly to *S*. Enteritidis, the rise in infections caused by other NTS serotypes may also suggest that there have been changes in the animal food production systems in the region during the last years, which were conducive to the formation of ecological niches suitable for the genetic and phenotypic traits of these serotypes. Presently, it is difficult to reveal the natural reservoirs of other NTS serotypes because these bacteria can persist in many different environments, and it is problematic to trace differential routes of transmission that result in human disease. Additionally, the frequency of the isolation of certain serotypes from the suspected food source reservoirs does not necessarily reflect the frequency of serotypes causing human infections [3]. 

For more detailed insights into the epidemiology of NTS in the region, we performed ERIC-PCR typing of our isolates. This technique provided a higher level of resolution compared to serotyping and MLST. The results of this analysis indicated a high degree of clonality among *S*. Typhimurium strains that have been frequently found in salmonellosis patients of the region during the last two decades and associated with a high level of MDR and ESBL production [22]. On the contrary, *S*. Enteritidis isolates constituted a more diverse group, with a greater variability in ERIC-PCR profiles. A number of NTS isolates in our collection resisted conventional agglutination tests but the majority of them produced ERIC-PCR profiles that were similar either to *S*. Typhimurium or *S*. Enteritidis profiles and therefore were placed together into the corresponding groups within these two serotypes. Thus, ERIC-PCR typing provides a better resolution and allows us to identify the strains that are difficult to type by conventional methods.

This study is the first WGS-based analysis of human NTS isolates from Armenia that provided a more in-depth assessment of the epidemiological situation with salmonellosis in the region as well as the identification of genetic traits associated with a high virulence potential. Genomic sequences helped to confirm the known serotypes, as well as to identify unknown serotypes, STs and the profiles of SPIs, plasmids, phages and virulence-related genes in our NTS isolates. All *S*. Enteritidis isolates in this study were assigned to ST11, regardless of their AMR phenotypes or year of isolation. Among the sequenced *S*. Typhimurium isolates, the most common ST was ST328, with 26 isolates that were collected from 1996 to 2016. Remarkably, 73.1% of *S*. Typhimurium ST328 isolates were ESBL-producers [22]. Moreover, the only bloodstream isolate in this study, *S*. Typhimurium A_684, also belonged to ST328 and was an ESBL-producer. The remaining four sequenced *S*. Typhimurium isolates belonged to ST19. Notably, the ST328 is a single-locus variant of ST19, which is one of the most globally spread STs of *S*. Typhimurium [42]. Contrary to the European and North American situation, however, we did not encounter any monophasic *S*. Typhimurium strains in our study. During the past two decades, the monophasic variant of *S*. Typhimurium with the antigenic formula 1,4,[5],12:i:- has spread globally, subsequently becoming one of the leading NTS infections in animals and humans [43]. This *S*. Typhimurium variant is linked to swine, and its absence in our collection may reflect the substantial absence of imports of pig products from the aforementioned countries.

The serotype of five isolates, the typing of which was not possible with conventional agglutination tests, was determined from their genomic sequences, as well as their STs. Of these, four MDR strains were identified as *S*. Derby ST40 (two isolates), *S*. Kentucky ST198, and *S*. Newport ST31, and a non-MDR isolate was identified as *S*. Agona ST13. It should be noted here that the four MDR isolates, which were not serotyped by agglutination tests, demonstrated serotype-specific AMR profiles differing from the profiles of all other MDR isolates in this study. Moreover, *S*. Derby ST40 isolates were resistant to fluoroquinolones but susceptible to β-lactams. These observations emphasise once more the importance of the correct and timely serotyping of NTS isolates to prescribe the most optimal antimicrobial treatment regimens.

From the genomic sequences of NTS isolates we identified virulence-related genes and the relevant genetic elements such as SPIs, virulence plasmids, and phages. The virulence-related factors and genes shared by all NTS serotypes were revealed, and serotype- and isolate-specific variations were characterised. Our results indicated that the main differences in the repertoire of identified virulence-associated factors were related mainly to a serotype and serotype-specific diversity of SPIs, virulence plasmids, and phage-encoded genes. The NTS virulence plasmids carrying *pef*, *svp*, *rck*, and *mig*-5 genes were identified in *S*. Enteritidis and *S*. Typhimurium isolates only, with a prevalence of 85.7% (6/7) and 13.3% (4/30), respectively. Thus, the high prevalence of NTS virulence plasmids was characteristic for *S*. Enteritidis isolates that have become the most common causative agent of salmonellosis in the region. Among *S*. Typhimurium isolates, the presence of *Salmonella* virulence plasmid was specific for ST19 isolates, whereas in ST328 isolates, circulating in the region for two decades and associated with a high level of MDR, the virulence-related plasmids were absent, irrespective of the year of isolation or AMR phenotype. This study also illustrated the presence of the HPI encoding a yersiniabactin-mediated iron acquisition system [31] in one human *S*. Typhimurium ST328 isolate from Armenia.

The diversity of virulence genes in different serotypes may offer some clues regarding possible phenotypic traits that shape a successful pathogen. For example, all our *S*. Enteritidis isolates, but not others, carried the *sefABCD* operon, which encodes genes involved in the production of SEF14 fimbriae (Table 1). The inactivation of the entire *sefABCD* operon in *S*. Enteritidis decreases its virulence in mice more than 1000-fold [44]. The important role in infection is played by the adhesion subunit of SEF14, SefD, since the *sefD* mutants are not easily internalised by peritoneal macrophages. This adhesion subunit, therefore, is crucial for the efficient uptake or survival of *S*. Enteritidis in macrophages [44]. Paradoxically, the lack of SefD increases the virulence of *S*. Enteritidis in hens [45]. Moreover, the expression of *sefD* has a protective effect and mitigates the disease, because birds infected with the wild type strain demonstrate decreased mortality compared to those infected with the *sefD* mutant. Thus, the presence of the *sefABCD* virulence operon in *S*. Enteritidis exerts differential effects depending on the host infected, and this may explain the success of this pathogen, which colonises poultry without causing disease. Unfortunately, there has been no follow-up work, and presently we do not have any mechanistic explanation of how the factors contributing to the virulence of *S*. Enteritidis in humans such as *sefABCD* may simultaneously serve as a morbidity and mortality protection mechanism in other animals such as poultry. Deciphering molecular mechanisms of this interaction could provide a much better understanding of the epidemiology of this important pathogen, including how it replaced *S*. Gallinarum in chicken [37] as a more successful coloniser and how it became an eminent pathogen on the global scale [41] being disseminated to many parts of the world. Better understanding of colonisation and maintenance mechanisms of this pathogen in the poultry host is also important for designing control and eradication measures. This approach is also applicable to other NTS bacteria that have reservoirs in agricultural animals.

In summary, the results of our study suggested several important conclusions regarding the epidemiology of salmonellosis in Armenia during the past two decades. During this period, NTS emerged as a leading cause of food-borne gastrointestinal infections in the region, with a significantly higher proportion of younger patients among the hospitalised patients. An in-depth analysis of NTS isolates indicated the clonal structure of *S*. Typhimurium populations, which were frequently multidrug-resistant. We did not detect monophasic *S*. Typhimurium strains among our isolates, which is different from the European and North American perspective, where this variant has become a leading cause of NTS infections during the past two decades. We also noted a gradual increase in the frequency of isolation of *S*. Enteritidis and other NTS serotypes during the study period, but with the opposite trend for *S*. Typhimurium and *S*. Arizonae. Genomic analyses revealed that although several virulence and virulence-associated genetic elements were present in all NTS isolates, most of them displayed the serotype-specific diversity of virulence genes, SPIs, virulence plasmids, and phages. Our results demonstrated the variability and dynamics in the epidemiology of salmonellosis in the area, with a high virulence potential present in human NTS isolates. Further investigations are necessary to uncover local reservoirs of NTS and transmission routes to ease the burden of the disease in the region.

## 4. Materials and Methods

### 4.1. Human Isolates of NTS

The study was carried out based on a collection of 316 isolates of NTS recovered from patients with salmonellosis admitted to the Nork Infectious Clinical Hospital (Ministry of Health, Armenia) between 1996 and 2019. Diagnosis was based on clinical presentations and laboratory analyses. Clinical presentations consistent with gastroenteritis were diarrhoea, fever, nausea, vomiting, and abdominal cramps. Presumptive *Salmonella* isolates were identified by performing standard biochemical tests including fermentation of glucose, negative urease reaction, lysine decarboxylase, negative indole test, H_2_S production, and fermentation of galactitol (dulcitol) [46]. Serotypes of isolates were determined in accordance with the White–Kauffmann–Le Minor scheme [47]. All available and eligible salmonellosis cases in the hospital in 2016 were included in the study, whereas sampling in other years was intermittent. The database of patients from 1996 to 2011 as well as the information on the serotypes of NTS were available only to a limited extent.

The study protocol was approved by the Ethics Committee of the Institute of Molecular Biology NAS RA (IORG number 0003427, Assurance number FWA00015042, and IRB number 00004079). Patients enrolled in the 2016–2019 study also provided written consent. Parental/guardian permission and consent were obtained when children/minors were enrolled in this study.

### 4.2. Antimicrobial Susceptibility Testing

NTS isolates were tested for susceptibility towards 14 AMs belonging to 10 different classes. The SOPs were strictly followed, in accordance with the guidelines of the Clinical and Laboratory Standards Institute (CLSI) for standard disk diffusion assays [48]. Muller–Hinton agar (Liofilchem^®^ s.r.l., Roseto degli Abruzzi, Italy) was used. Bacterial inoculum was adjusted to the equivalent of a 0.5 McFarland standard. The following AM disks (Liofilchem^®^ s.r.l., Italy) were used: ampicillin (10 µg), amoxicillin-clavulanic acid (20 µg/10 µg), ceftazidime (30 µg), ceftriaxone (30 µg), chloramphenicol (30 µg), ciprofloxacin (5 µg), gentamicin (10 µg), imipenem (10 µg), nalidixic acid (30 µg), streptomycin (10 µg), sulphonamide (300 µg), tetracycline (30 µg), and trimethoprim-sulfamethoxazole (1.25 µg/23.75 µg). The results of susceptibility testing were interpreted based on the CLSI criteria [48]. The minimum inhibitory concentration (MIC) of azithromycin was determined by performing the agar dilution method according to the CLSI standards [49]. Isolates that showed resistance to representatives of at least three classes of AMs were considered as MDR [50]. The ESBL phenotype was identified by the double-disk test using ceftriaxone and ceftazidime with or without clavulanic acid, according to the guidelines of the CLSI [48]. *E. coli* strains ATCC 25922 and ATCC 35218 were used for quality control.

### 4.3. Bacterial DNA Extraction

Total bacterial DNA samples for ERIC-PCR analysis were isolated using the boiling lysate protocol [51] and frozen at −20 °C until the genotyping assays. 

For WGS, bacterial DNA samples were extracted using the UltraClean^®^ Microbial DNA Isolation Kit (MO BIO Laboratories Inc., San Diego, CA, USA) according to the manufacturer’s recommendations. DNA samples were stored in 10 mM Tris (with no EDTA) at −20 °C. 

### 4.4. ERIC-PCR Typing/Analysis

Among 316 human NTS isolates in our collection, 112 were selected for ERIC-PCR analysis based on the serotype, year of isolation and antimicrobial resistance (AMR) profile. The primers used for ERIC-PCR were: ERIC-1R (5′-ATGTAAGCTCCTGGGGATTCAC-3′) and ERIC-2 (5′-AAGTAAGTGACTGGGGTGAGCG-3′) (Integrated DNA Technologies, BVBA—Löwen, Belgium) [52]. PCR was performed as described previously [53] with some changes. Briefly, the PCR conditions included an initial denaturation at 94 °C for 4 min, followed by 35 cycles of denaturation at 94 °C for 1 min, primer annealing at 52 °C for 1 min, an extension at 72 °C for 4 min, and a final extension at 74 °C for 10 min. The amplified products were separated by performing gel electrophoresis in 1.5% agarose. HyperLadder™ 1 kb (Bioline, Memphis, AZ, USA) was used as a molecular weight marker.

The amplicon patterns generated by ERIC-PCR were analysed with the gel analysis software GelAnalyzer 19.1 (www.gelanalyzer.com, accessed on 15 May 2022). After normalisation and pattern alignment, the dendrogram showing the genetic similarity among isolates was generated using the Dice coefficient and the Unweighted Pair Group Method with Arithmetic Average (UPGMA) algorithm for cluster analysis (http://insilico.ehu.eus/dice_upgma/, accessed on 15 May 2022).

### 4.5. Whole Genome Sequencing of NTS Isolates

Whole genome sequencing of 42 NTS isolates in the study was provided by MicrobesNG (https://microbesng.com/, accessed on 15 May 2022), which was supported by the BBSRC (grant number BB/L024209/1). Sequencing was performed on the Illumina MiSeq and HiSeq 2500 platforms. Bacterial isolates were sequenced using 2 × 250 bp paired-end reads at 30x coverage or higher. Reads were adapter-trimmed using Trimmomatic 0.30 with a sliding window quality cut off Q15 [54]. The contigs were annotated using Prokka 1.11 [55]. Assembly metrics were calculated using QUAST [56]. Taxonomic rank assignment was carried out using the Kraken software [57]. Whole genome sequences of NTS isolates are available in the European Nucleotide Archive (ENA) database under Project PRJEB36290. Accession numbers for individual isolates are listed in Appendix A.

### 4.6. Bioinformatics Analyses

The general information on the genomes of NTS isolates and genomic components was obtained using Genomics tools of the Bacterial and Viral Bioinformatics Resource Center (BV-BRC, https://www.bv-brc.org, accessed on 20 June 2022). Serotypes of NTS strains were identified using a web-based tool SeqSero2 [25,26] (www.denglab.info/SeqSero2, accessed on 22 July 2022). The assignment of isolates to STs was performed using a PubMLST databases [27] (https://pubmlst.org, accessed on 20 June 2022). In silico prediction of known or potential virulence factors was performed using the Virulence Factor Database (VFDB) [28] (http://www.mgc.ac.cn/cgi-bin/VFs/v5/main.cgi, accessed on 20 June 2022) and Pathosystems Resource Integration Center (PATRIC) [29] (https://www.patricbrc.org, accessed on 20 June 2022) web resources. The online search tool SPIFinder [30] (https://cge.cbs.dtu.dk/services/SPIFinder, accessed on 15 June 2022) was used for the prediction of *Salmonella* pathogenicity islands. The Average Nucleotide Identity (ANI) value was determined using the ANI Calculator tool [32] (https://www.ezbiocloud.net/tools/ani, accessed on 20 June 2022). PlasmidFinder 2.1 tool [33] (https://cge.cbs.dtu.dk/services/PlasmidFinder, accessed on 20 June 2022) was used to predict plasmids. The annotation of prophage sequences within bacterial genomes was performed using the PHASTER web server [34,35] (https://phaster.ca, accessed on 20 June 2022). Contigs were analysed using BLAST (http://blast.ncbi.nlm.nih.gov/Blast.cgi, accessed on 15 May 2022).

### 4.7. Statistical Analyses

The *p* Value (two-tailed) from Fisher’s exact test was calculated using the on-line GraphPad QuickCalcs resource (http://www.graphpad.com/quickcalcs/contingency1.cfm, accessed on 15 May 2022) to evaluate statistical differences between the compared groups. *p* Values ≤ 0.05 were considered significant.

## Figures and Tables

**Figure 1 ijms-23-09330-f001:**
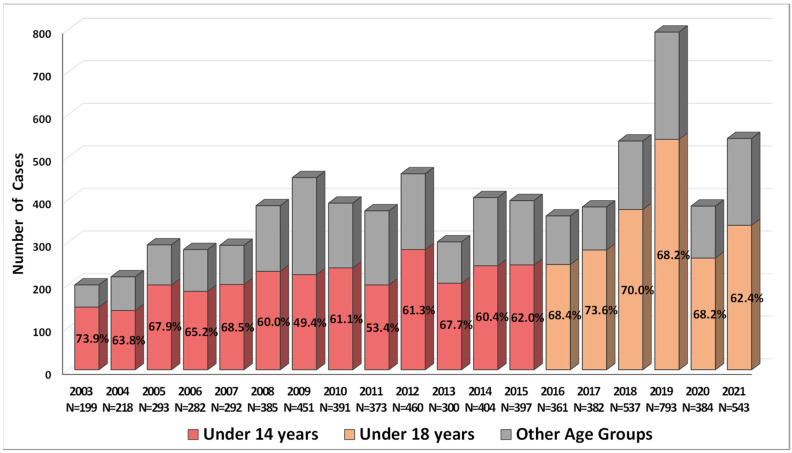
The total number of confirmed non-typhoidal *Salmonella* infections in Armenia and the proportion of age groups under 14 and 18 years old, according to the Statistical Committee of the Republic of Armenia data [23].

**Figure 2 ijms-23-09330-f002:**
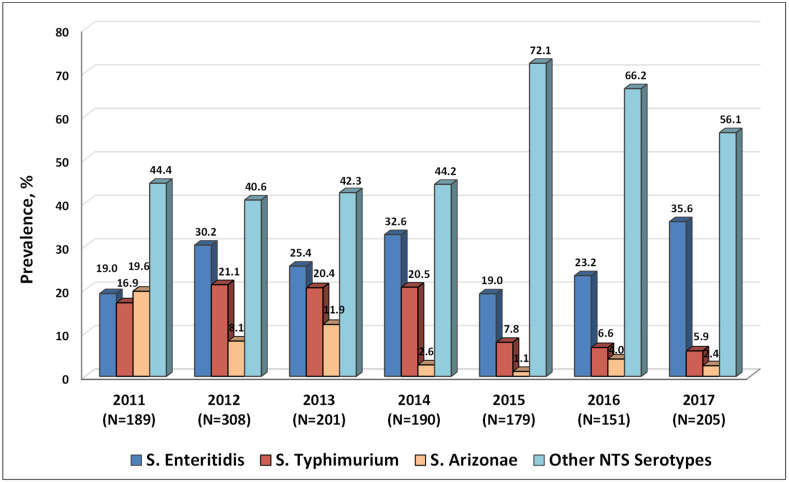
Serotypes of 1423 human non-typhoidal *Salmonella* (NTS) isolates recovered from patients admitted to the Nork Infectious Clinical Hospital (Ministry of Health, Armenia) in 2011–2017.

**Figure 3 ijms-23-09330-f003:**
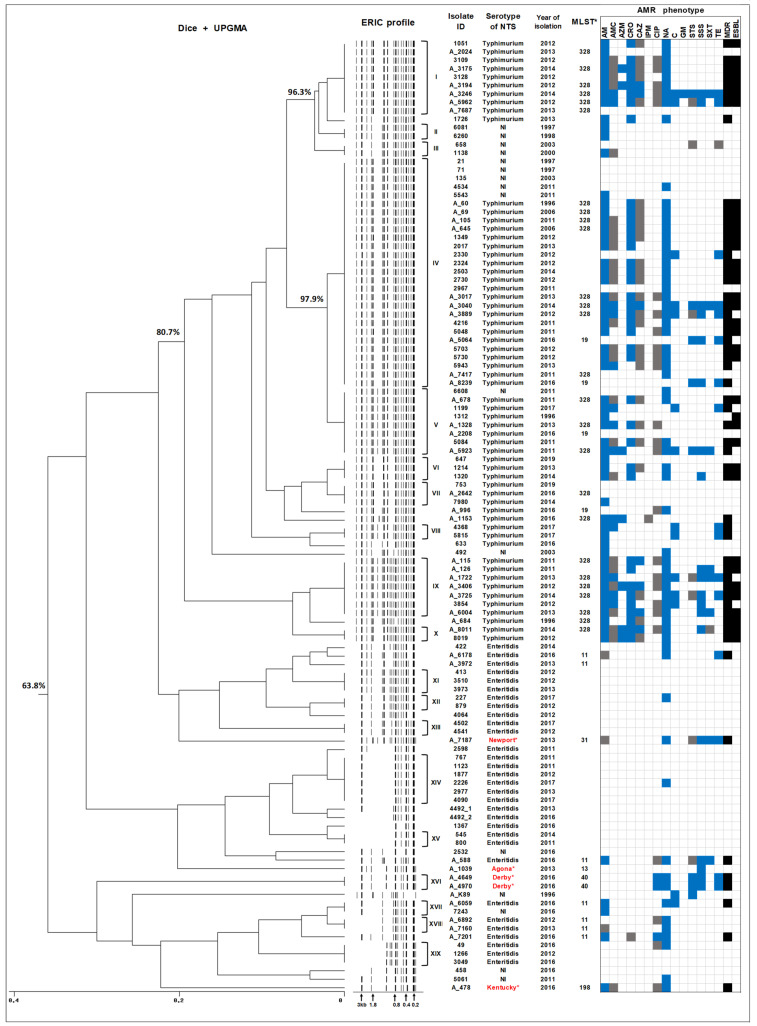
Dendrogram of ERIC-PCR fingerprinting profiles of 112 human NTS isolates from Armenia, with sequence types (MLST) and AMR phenotypes. The dendrogram was generated with Dice coefficient and the UPGMA clustering method. Colour keys: blue nodes represent resistant phenotype to antimicrobial agent, grey nodes represent intermediate susceptibility to antimicrobial agent, black nodes represent positive isolates for MDR or ESBL phenotype. Abbreviations: ERIC, Enterobacterial Repetitive Intergenic Consensus; NTS, non-typhoidal *Salmonella*; AMR, antimicrobial resistance; AM, ampicillin; AMC, amoxicillin-clavulanic acid; AZM, azithromycin; CRO, ceftriaxone; CAZ, ceftazidime; IPM, imipenem; CIP, ciprofloxacin; NA, nalidixic acid; C, chloramphenicol; GM, gentamicin; STS, streptomycin; SSS, sulfonamide; STX, trimethoprim-sulfamethoxazole; TE, tetracycline; MDR, multidrug resistant; ESBL, extended spectrum beta-lactamases. *—Identified based on WGS data.

**Figure 4 ijms-23-09330-f004:**
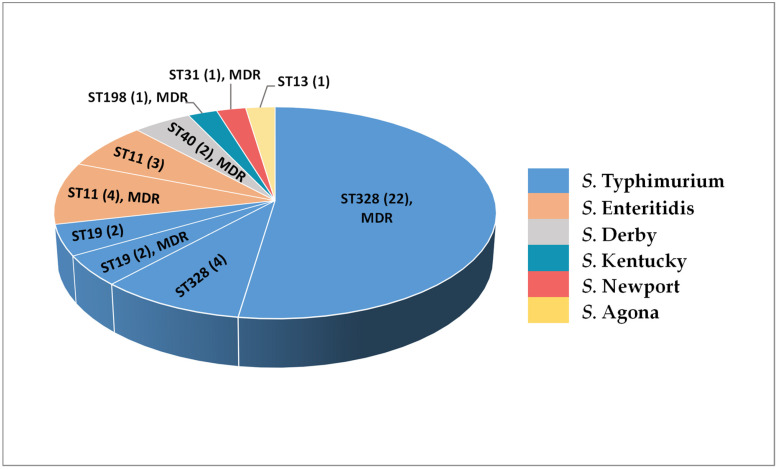
Serotypes and sequence types (ST) of 42 human non-typhoidal *Salmonella* (NTS) isolates from Armenia determined by whole genome sequencing (WGS). MDR, all phenotypically multidrug-resistant.

**Table 1 ijms-23-09330-t001:** Prevalence of the serotype-specific virulence-related genes in human non-typhoidal *Salmonella* (NTS) isolates from Armenia identified by VFanalyzer [28].

Genes	Serotype of NTS (Number of Isolates)						Virulence Factor Class
	Typhimurium (*n* = 30)	Enteritidis (*n* = 7)	Derby (*n* = 2)	Agona (*n* = 1)	Kentucky (*n* = 1)	Newport (*n* = 1)	
*lpfABCDE*	-	7	-	1	1	1	Fimbrial adherence determinants
*peg*	-	7	2	-	-	1	
*pefABCD*	4	6	-	-	-	-	
*safA*	30	7	2	-	1	1	
*sefABCD*	-	7	-	-	-	-	
*staABCDEFG*	-	-	2	1	-	-	
*stcA*	30	-	-	1	-	-	
*stcB*	30	-	-	1	1	-	
*stcC*	30	-	-	1	1	-	
*stcD*	30	-	-	1	-	-	
*steABCDEF*	-	7	2	1	1	1	
*stjBC*	30	-	-	1	1	1	
*stkABCDEFG*	-	-	-	-	1	-	
*tcfABCD*	-	-	-	-	1	-	
*ratB*	30	7	-	1	-	1	Nonfimbrial adherence determinants
*sopE*	-	7	-	-	-	1	T3SS1 translocated effector
*gogB*	30	-	-	-	-	-	T3SS2 translocated effector
*sopD2*	-	7	2	1	1	1	
*spvCD*	4	6	-	-	-	-	
*sseI/srfH*	30	7	-	-	-	-	
*sseK1*	30	7	2	1	1	-	
*sseK2*	30	7	2	-	1	1	Macrophage inducible gene
*mig-5*	4	6	-	-	-	-	
*sodC1*	30	7	-	-	-	-	Stress adaptation
*rck*	4	6	-	-	-	-	Serum resistance
*spvB*	4	6	-	-	-	-	Toxin
*upaG/ehaG*	-	-	2	-	-	-	Autotransporter
*ehaB*	-	-	-	-	1	-	
*faeCDEHIJ*	-	-	-	-	1	-	Adherence/K88 fimbriae (*Escherichia*)

## Data Availability

Data associated with this article are included in the Appendix A. Whole genome sequences of NTS isolates are available in the European Nucleotide Archive (ENA) database under Project PRJEB36290. Accession numbers for individual isolates are listed in the Appendix A.

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
