# Peer review of "Molecular Epidemiology and Virulence of Non-Typhoidal *Salmonella* in Armenia"

_ijms, 2022, doi:10.3390/ijms23169330_

Round 1

Reviewer 1 Report

Taken together this is a fairly well written work describing Salmonella isolates in Armenia over the past 20 years.

I only have a number of minor requests:

1) The picture described is quite different from the European and North American one which in recent years has been characterized by the spread of monophasic Salmonella, of which there seems to be no trace in Armenia. I would emphasize this aspect under discussion. I guess this indicates a substantial absence of imports of poultry or pig products from the aforementioned countries.

2) the phrase in the introduction "the establishment of the epidemiological link between NTS strains in food-producing animals and the human disease is not straightforward" does not convince me. The fact that the factors contributing to NTS survival in the food chain are not fully understood cannot question the fact that the main source of human contamination are products of animal origin.

3) the statement under discussion that S. Enteritidis has replaced S. Gallinarum does not seem correct to me. S. Enteritidis is a serotype capable of colonizing poultry without causing disease, while S. Gallinarum is a disease inducing species in chickens.

4) Line 357: please specify - The OXIDATIVE stress adaptation virulence factor sodC1

Author Response

Reviewer 1

Taken together this is a fairly well written work describing Salmonella isolates in Armenia over the past 20 years.

Response: Thank you for your overall positive evaluation.

I only have a number of minor requests:

1) The picture described is quite different from the European and North American one which in recent years has been characterized by the spread of monophasic Salmonella, of which there seems to be no trace in Armenia. I would emphasize this aspect under discussion. I guess this indicates a substantial absence of imports of poultry or pig products from the aforementioned countries.

Response:

Thank you very much indeed for your constructive comment. Indeed, we have not seen any monophasic S. Typhimurium among our sequenced isolates. To emphasise this difference from the European and North American situation, we added the full antigenic/phase formulae to Table S1, added a brief description to Results, and discussed this difference in the discussion section. These text additions are highlighted in yellow in the revision.

2) the phrase in the introduction "the establishment of the epidemiological link between NTS strains in food-producing animals and the human disease is not straightforward" does not convince me. The fact that the factors contributing to NTS survival in the food chain are not fully understood cannot question the fact that the main source of human contamination are products of animal origin.

Response:

We appreciate your comment. While indeed the majority of cases can be tracked back to the animal origin, the numbers of Salmonella contaminations of salads, peanut butter, chocolate products, etc. are on the rise. And this kind of contaminations are difficult to track back to the animal origin. We agree, however, that the sentence should be less categorical. Thus, we changed the sentence to “However, uncovering the epidemiological link between NTS strains in food-producing animals and the human disease is not always straightforward.” 

3) the statement under discussion that S. Enteritidis has replaced S. Gallinarum does not seem correct to me. S. Enteritidis is a serotype capable of colonizing poultry without causing disease, while S. Gallinarum is a disease inducing species in chickens.

Response:

We wrote in Discussion that “It has been observed that targeted elimination of S. Gallinarum in chicken in the USA led to the parallel sudden increase in S. Enteritidis, which may persist in chicken without clinical signs of disease [36]”. This is not our statement but a referral to a work that has been published in a well-respected peer-reviewed academic journal: 

Baumler, A.J.; Hargis, B.M.; Tsolis, R.M. Tracing the origins of Salmonella outbreaks. Science 2000, 287, 50–52. doi: 10.1126/science.287.5450.50.

4) Line 357: please specify - The OXIDATIVE stress adaptation virulence factor sodC1

Response: 

Thank you, the word “oxidative" has been added as suggested.

Reviewer 2 Report

1. Introduction is too lengthy and should be shortened considerably. The authors should only include information relevant to the current study, e.g. ecology of salmonellosis in Armenia. The section should be finished by clear aims of the study.

2. Results:

The entire section is too lengthy and poorly organised. As the authors characterise different serotypes of Salmonella, the arguments often jump between different serotypes, STs or clusters. The authors must re-organise the entire section and report the WGS-based results according to the serotype and NOT the genetic element analysed. Moreover, the authors need to focus on most significant findings only as the detailed results are already present in form of various supplementary tables. It is crucial that the structure is improved as in the current form it is difficult to read through the manuscript and the main points might be missed by the reader.

Specific points:

Incidence of salmonellosis in Armenia in 2003-2021 - this paragraph contains information that are not a result of the current study and should be moved to introduction section.

Serotypes of NTS circulating in Armenia in 2011-2017 - lines 164-174 - again this should be moved to either introduction or discussion. The Results section should only contain the actual results of this study.

ERIC-PCR Subtyping of human NTS isolates circulating in Armenia - the method of selection of the 'representative' isolates should be moved to Materials and Methods

Figure 3 - please change the antimicrobial drug abbreviations according to an established international system such as EUCAST; - if ESBL genes were identified using WGS but not confirmed using phenotypic testing, they cannot be indicated under 'AMR phenotype' 

Figure 4 - this figure should be replaced by a table which in addition to the data included in the piechart should contain a list of identified AMR genes and identified plasmids as the AMR genes are often associated with mobile elements. Consequently, a section on genetic determinants should be added and possibly combined with  the section on detected plasmids.

Discussion:

Again, here the authors should focus on specific blocks of information and focus on most interesting/significant observations. These results should be then discussed in the context of the current literature and/or epidemiology of salmonellosis in Armenia. E.g. as  the study does not examine virulence of Salmonella, the authors do not need to discuss possible effect of specific virulence factors on the isolates characterised in this study.

To make the discussion more concise, the authors should answer the question: what is the main conclusion of the study and why is it important?

This can be followed by: Based on the results what is the current epidemiology of human salmonellosis in Armenia? Did it change over the years? How and why?

What is the main finding about AMR? Is there an increase/decrease of resistance to specific antimicrobials? Is it the same for all serotypes? Why?

Supplementary materials:

There seems to be a highly inflated number of AMR genes identified using CARD and PATRIC db (between 50 and 70??) Please correct the error.

Where no serotype was determined please add the antigenic formula including the missing antigen/phase.

Etc.

Any considerations about the strengths and weaknesses of the study should be added.

Author Response

Reviewer 2

1. Introduction is too lengthy and should be shortened considerably. The authors should only include information relevant to the current study, e.g. ecology of salmonellosis in Armenia. The section should be finished by clear aims of the study.

Response: 

In scientific articles, the introduction part serves several purposes. It should demonstrate what is known on the subject and describe the current state-of-the-art in the field. Further, it is aimed at identifying the existing knowledge gaps in the field and place own research within the established body of knowledge. 

Besides, NTS is a pathogen of global importance, not an endemic pathogen confined to a certain geographical location. Moreover, it displays differential epidemiology in different parts of the world, with rapidly changing dynamic. Because of this, the introduction section in every single paper on Salmonella describes the current international situation with it to place own research within an international context and generate a meaningful comparison with the existing data.

Thus, limiting Introduction to the ecology of salmonellosis in Armenia would severely limit  this part of paper to the activity of a single research group working in this area in the country, that is us. And this would certainly limit the integration of our research into the existing body of knowledge and take it out from the existing framework of international research activities dealing with the epidemiological and other aspects of this global pathogen.

Regarding the aims: the last paragraph in Introduction states the aims of this study.

Hence, we believe the introduction section should follow the commonly accepted standards exemplified by many publications in the field.

2. Results:

The entire section is too lengthy and poorly organised. As the authors characterise different serotypes of Salmonella, the arguments often jump between different serotypes, STs or clusters. The authors must re-organise the entire section and report the WGS-based results according to the serotype and NOT the genetic element analysed. Moreover, the authors need to focus on most significant findings only as the detailed results are already present in form of various supplementary tables. It is crucial that the structure is improved as in the current form it is difficult to read through the manuscript and the main points might be missed by the reader.

Response:

First of all, the number of isolates within each serotype is very uneven, which corresponds to the frequency of serotypes circulating in this geographical location. Correspondingly, the range of WGS data varies from the 30 genomes of S. Typhimurium to the single genomes of S. Agona, S. Kentucky, and S. Newport. Thus, presenting data in a serotype-centric format instead of genocentric would results in a highly imbalanced structure of Results. Second, the suggested format would lack a comparative aspect of this research. Thus, if to follow the serotype-centric format, it would be necessary to add a new section, devoted to the comparative analyses of NTS serotypes in terms of genetic elements identified. This would inevitably lead to the reiteration of genetic data that are already presented for each serotype separately and unnecessarily escalate the volume and complexity of Results.

The format of presentation of our work follows the generally accepted formats in the field, and we would like to adhere to this commonly accepted approach. 

Specific points:

Incidence of salmonellosis in Armenia in 2003-2021 - this paragraph contains information that are not a result of the current study and should be moved to introduction section.

Response:

Thank you for the comment. It should be clarified here that our reference to the Statistical Committee data is the reference to raw data. In the current work, these data were subjected to vigorous statistical analyses to reveal several important features. First, the variability in the total number of confirmed cases per year was evaluated, with median 384 (interquartile range (IQR) 293-451, range 199-793). Second, a significantly higher incidence rate in the younger age groups compared with the remaining age groups combined was established. These findings are the results of the current study, and they have not been published before. Therefore, they cannot be moved to Introduction.

Serotypes of NTS circulating in Armenia in 2011-2017 - lines 164-174 - again this should be moved to either introduction or discussion. The Results section should only contain the actual results of this study.

Response: 

In science, it is not a rare occasion that the results collected are analysed at a later time. This is especially true for epidemiological studies that may encompass many years of sample collection. In this particular case, these are the results that have been collected before but analysed in the current study. The analysis performed allowed to identify the most common serotypes causing the disease as well as to identify the main trends in serotype distribution. Moreover, we revealed the gradual increase in S. Enteritidis infections in Armenia. We summarised these results in Fig. 2. Since these are our current analytical results and they have not been published before, they cannot be moved somewhere else.

ERIC-PCR Subtyping of human NTS isolates circulating in Armenia - the method of selection of the 'representative' isolates should be moved to Materials and Methods

Response:

The logic for the placement of selection criteria for ERIC-PCR typing are based on the context of results presented in the two previous sections, exemplified by Figures 1 and 2. The proposed separation of these criteria will leave the criteria without supportive background information.

Figure 3 - please change the antimicrobial drug abbreviations according to an established international system such as EUCAST; - if ESBL genes were identified using WGS but not confirmed using phenotypic testing, they cannot be indicated under 'AMR phenotype' 

Response:

Susceptibility to antimicrobials was tested in accordance with the guidelines of the Clinical and Laboratory Standards Institute (CLSI, 28th ed. [46]) as described in Materials and Methods. All the drug abbreviations used are from the Glossary II, pp. 248-252. 

In Figure 3, only AMR phenotypes are shown, without the involvement of any WGS data, and this is clearly specified in the figure legend.

MLST and serotyping data that were produced with the use of WGS data are specified in the footnote as “ * - Identified based on WGS data”. Thus, no AMR data in this figure are generated with the use of WGS data.

Figure 4 - this figure should be replaced by a table which in addition to the data included in the piechart should contain a list of identified AMR genes and identified plasmids as the AMR genes are often associated with mobile elements. Consequently, a section on genetic determinants should be added and possibly combined with  the section on detected plasmids.

Response:

Extensive analyses of AMR genes as well as the mobile genetic elements associated with AMR in our NTS isolates have been published before:

Sedrakyan, A.M.; Ktsoyan, Z.A.; Arakelova, K.A.; Zakharyan, M.K.; Hovhannisyan, A.I.; Gevorgyan, Z.U.; Mnatsakanyan, A.A.; Kakabadze, E.G.; Makalatia, K.B.; Chanishvili, N.A.; Pirnay, J.-P.; Arakelyan, A.A.; Aminov, R.I. Extended-Spectrum β-Lactamases in Human Isolates of Multidrug-Resistant Non-typhoidal Salmonella enterica. Front. Microbiol. 2020, 11, 592223. doi: 10.3389/fmicb.2020.592223.

This work was cited on several occasions in the present study, but we do not feel it would be appropriate to include the data, which have been already published, in the current study. Besides, antimicrobial resistance is not the focus of this study.

Discussion:

Again, here the authors should focus on specific blocks of information and focus on most interesting/significant observations. These results should be then discussed in the context of the current literature and/or epidemiology of salmonellosis in Armenia. E.g. as  the study does not examine virulence of Salmonella, the authors do not need to discuss possible effect of specific virulence factors on the isolates characterised in this study.

To make the discussion more concise, the authors should answer the question: what is the main conclusion of the study and why is it important?

Response:

The main conclusions of this study are summarised in Abstract. 

This can be followed by: Based on the results what is the current epidemiology of human salmonellosis in Armenia? Did it change over the years? How and why?

Response:

These questions are extensively discussed in Discussion and summarised in Abstract.

What is the main finding about AMR? Is there an increase/decrease of resistance to specific antimicrobials? Is it the same for all serotypes? Why?

Response:

We performed very extensive research on AMR among our NTS isolates, which has been published in 2020 and which is cited on several occasions here:

Sedrakyan, A.M.; Ktsoyan, Z.A.; Arakelova, K.A.; Zakharyan, M.K.; Hovhannisyan, A.I.; Gevorgyan, Z.U.; Mnatsakanyan, A.A.; Kakabadze, E.G.; Makalatia, K.B.; Chanishvili, N.A.; Pirnay, J.-P.; Arakelyan, A.A.; Aminov, R.I. Extended-Spectrum β-Lactamases in Human Isolates of Multidrug-Resistant Non-typhoidal Salmonella enterica. Front. Microbiol. 2020, 11, 592223. doi: 10.3389/fmicb.2020.592223.

All these questions are addressed in the above publication in great detail. This is an open access publication and thus can be accessed by anyone interested in the topic, even without access to academic libraries.

Supplementary materials:

There seems to be a highly inflated number of AMR genes identified using CARD and PATRIC db (between 50 and 70??) Please correct the error.

Response:

CARD and PATRIC databases are extensively used by the research community around the globe, and the results of identification of AMR genes with these databases have been reported in numerous peer-reviewed publications. Results obtained with these databases are the results, which should be reported. They cannot be altered as suggested.

Where no serotype was determined please add the antigenic formula including the missing antigen/phase.

Response:

The antigenic formulae were added to Table S1 as requested. 

Etc.

Any considerations about the strengths and weaknesses of the study should be added.

Response:

Strengths and weaknesses of the study are discussed throughout on multiple occasions. For example, we discussed the strength of WGS approach that allowed to identify the serotypes that resisted identification using the  classical agglutination tests. Or, on the weakness side, the inability to uncover the animal reservoirs of NTS and the routes of transmission to humans was discussed.  

Round 2

Reviewer 2 Report

I would like to thank the authors for the detailed responses to the suggestions. Unfortunately, it seems that the authors did not fully understand the revisions that were requested and did not make the changes requested. Therefore the manuscript still needs major revisions before it can be accepted for publication. 

Specific changes that must be introduced are listed below:

1. Introduction - Introduction should be relevant to the results described in the manuscript. This study analyses a set of strains isolated from hospitalised patients and uses WGS tools to detect AMR genes, virulence related genes, CRISPR-cas related elements, phages, etc. This is not work that studies virulence of specific strains. Especially since hospitalisation (and in particular of children) cannot be used as a equivalent  to increased virulence. There are great reviews already published on the virulence of Salmonella and therefore the introduction should not contain 4 paragraphs that treat  virulence determinants. In the interest of the readers, please shorten this sections to 1 paragraph and provide appropriate review references for the readers.

The authors are correct stating that the introduction should mention current gaps in a specific field of research but only in  the area that is relevant to the current study. In this case no gaps about virulence have been filled by this work and therefore focusing the introduction on this area is not appropriate.

The authors are also correct stating that : the introduction section in every single paper on Salmonella describes the current international situation with it to place own research within an international context and generate a meaningful comparison with the existing data. Thus, limiting Introduction to the ecology of salmonellosis in Armenia would severely limit this part of paper to the activity of a single research group working in this area in the country, that is us. And this would certainly limit the integration of our research into the existing body of knowledge and take it out from the existing framework of international research activities dealing with the epidemiological and other aspects of this global pathogen.

Therefore the authors must add this information to the introduction section, as it is currently missing. To be precise, please include details on ecology of salmonellosis in Armenia, information about specific noted outbreaks in the recent years that could explain the growing trends in the incidence of sepecific NTS in given years. Please describe the current policies that the coutry have that could explain why some NTS decreased - e.g. culling of breeders of laying hens positive for specific serotypes (name the serotypes), add information about the trade with other countries such as EU that had experienced increase in Enteritidis in current years, due to infection of lying hens and export of contaminated eggs. For the convenice of the authors, some of this arguments can be included in discussion section.

2. On contrary to the authors opinion, the aims are not clearly described and must be improved. The standard scheme "the aims of the study were ..." can be used at the end of the paragraph for clarity. In the current form, the last paragraph describes only what the authors did, but not to what end. 

3. L 131-134 The authors state that WGS has the highest typing resolution, yet they decided to use |PCR based method rather than cgMLST or SNP typing. Did the authors use high resolution WGS clustering method for the 42 analysed strains? If not, then please remove the sentence as it suggests that WGS based typing (other than MLST) was used in the study.

4. Paragraph L164-174 - must be removed or included in the introduction. THese are not results of the study but the summary of several studies already published and as such is a good paragraph for the introduction. 

5. Paragraph L 196 - L203 - Although it can be accepted to leave an introductory sentence in the results sections, this part must be moved to methods. The materials and methods section must be by itself sufficient to replicate the study and therefore the method of selection of strains should be clearly and in detail described in Materials and Methods section. If prefered the authors can leave this paragraph, but add more detailed description of selection strategy in MM section. 

6. The abbreviations must be changed. The authors are welcome to use CLSI method of naming antimicrobials, i.e. use full names instead of abbreviations as using full names is the scheme adapted by CLSI to avoid misinterpretations due to many different abbreviations currently found for diagnostic products. The authors cannot pick and choose abbreviations from the glossary that states ALL abbreviations for diagnostic and manufactured products currently or in the past on the market. In fact abbreviation such as AUG that the authors used for amoxicillin-clavulanic acid is not acceptable as it stands for Augmentin, which is a trade name of a drug manufactured by GSK. Please change into accepted standard abbreviations or use abbreviations of generic compounds throughout the manuscript and in supplementary methods. 

7. L256 - The fact that strains cluster together cannot be used to determine their serotype, please amend the sentence accordingly.

8. Paragraph L274 - 284 - Again please move this part to MM section, these are not results. 

Figure 4 - this figure is not needed and it is confusing. Please remove or move to supplementary information. This information is already found in the figure 3 and doesn't need to be repeated.

9. Considering high number of AMR genes detected, please add supplementary table with the list of the genes, and the percentage identity and coverage. These unusual results have to be made available for revision and for the readers. 

10. L515-L519 - The authors did not analyse the animal reservoirs but only human strains, how can this be considered a weakness of the study then?

11.L 559- 561 - serotype determination is not needed to antimicrobial resistance testing. It is sufficient to identify the strain to the species level to treat the patient according to the phenotypic susceptibility tests. Please remove the sentence. 

12. Paragraph L673 - please add parameters used for Prokka and the QC criteria adapted for quast and kraken analysis. Which thresholds were adopted to remove poor quality strains?

13. SeqSero has since a few years been superseded by \SeqSero2 which is more accurate and updated version, why did the authors use the old version? Please add whether reads or assemblies were used for the analysis. 

14. L701 What was Blast used for, which blast? Blast nucleotide? Which parameters were used? Please add this information.

15. considering the high number of pargraphs, please add conclusion section adding the most important conclusions of the study. It is not sufficient that it can be found in the abstract. 

Author Response

Responses to reviewer 2

1. Introduction - Introduction should be relevant to the results described in the manuscript. This study analyses a set of strains isolated from hospitalised patients and uses WGS tools to detect AMR genes, virulence related genes, CRISPR-cas related elements, phages, etc. This is not work that studies virulence of specific strains. Especially since hospitalisation (and in particular of children) cannot be used as a equivalent  to increased virulence. There are great reviews already published on the virulence of Salmonella and therefore the introduction should not contain 4 paragraphs that treat  virulence determinants. In the interest of the readers, please shorten this sections to 1 paragraph and provide appropriate review references for the readers.

The authors are correct stating that the introduction should mention current gaps in a specific field of research but only in  the area that is relevant to the current study. In this case no gaps about virulence have been filled by this work and therefore focusing the introduction on this area is not appropriate.

Response: We made the next changes to accommodate these requests:

Since AMR genes and CRISPR-Cas elements are not relevant to the scope of this article, the corresponding columns in Table S1 were removed in the revision.

The scope of the current paper is revealing genetic basis for virulence in our NTS isolates. Thus, the topic of virulence should be introduced to readers as a necessary background information.

The virulence potential of NTS isolates in the region have not been described before. Thus we believe we made a significant contribution to this field. 

The authors are also correct stating that : the introduction section in every single paper on Salmonella describes the current international situation with it to place own research within an international context and generate a meaningful comparison with the existing data. Thus, limiting Introduction to the ecology of salmonellosis in Armenia would severely limit this part of paper to the activity of a single research group working in this area in the country, that is us. And this would certainly limit the integration of our research into the existing body of knowledge and take it out from the existing framework of international research activities dealing with the epidemiological and other aspects of this global pathogen.

Therefore the authors must add this information to the introduction section, as it is currently missing. To be precise, please include details on ecology of salmonellosis in Armenia, information about specific noted outbreaks in the recent years that could explain the growing trends in the incidence of sepecific NTS in given years. Please describe the current policies that the coutry have that could explain why some NTS decreased - e.g. culling of breeders of laying hens positive for specific serotypes (name the serotypes), add information about the trade with other countries such as EU that had experienced increase in Enteritidis in current years, due to infection of lying hens and export of contaminated eggs. For the convenice of the authors, some of this arguments can be included in discussion section.

Response: We extended the last paragraph into the two to include the local information regarding human isolates of Salmonella. Regarding other aspects suggested: the focus of this paper is molecular epidemiology and virulence potential of human NTS isolates. It is beyond the scope of this paper to discuss the epidemiology of Salmonella in poultry, governmental policies and regulations, agricultural practices, trades with other countries and other irrelevant issues.

2. On contrary to the authors opinion, the aims are not clearly described and must be improved. The standard scheme "the aims of the study were ..." can be used at the end of the paragraph for clarity. In the current form, the last paragraph describes only what the authors did, but not to what end.

Response:

As requested, the last paragraph in Introduction starts with "The aims of the present study were ..."

3. L 131-134 The authors state that WGS has the highest typing resolution, yet they decided to use |PCR based method rather than cgMLST or SNP typing. Did the authors use high resolution WGS clustering method for the 42 analysed strains? If not, then please remove the sentence as it suggests that WGS based typing (other than MLST) was used in the study.

Response:

This sentence was a reference to the changing methodological approach that is taking place in the Salmonella research community. It is removed in the revision.

4. Paragraph L164-174 - must be removed or included in the introduction. THese are not results of the study but the summary of several studies already published and as such is a good paragraph for the introduction.

Response:

As requested, this paragraph was moved to Introduction (highlighted in yellow).

5. Paragraph L 196 - L203 - Although it can be accepted to leave an introductory sentence in the results sections, this part must be moved to methods. The materials and methods section must be by itself sufficient to replicate the study and therefore the method of selection of strains should be clearly and in detail described in Materials and Methods section. If prefered the authors can leave this paragraph, but add more detailed description of selection strategy in MM section. 

Response:

As requested, this information is added to section 4.4 (highlighted in yellow). 

6. The abbreviations must be changed. The authors are welcome to use CLSI method of naming antimicrobials, i.e. use full names instead of abbreviations as using full names is the scheme adapted by CLSI to avoid misinterpretations due to many different abbreviations currently found for diagnostic products. The authors cannot pick and choose abbreviations from the glossary that states ALL abbreviations for diagnostic and manufactured products currently or in the past on the market. In fact abbreviation such as AUG that the authors used for amoxicillin-clavulanic acid is not acceptable as it stands for Augmentin, which is a trade name of a drug manufactured by GSK. Please change into accepted standard abbreviations or use abbreviations of generic compounds throughout the manuscript and in supplementary methods.

Response: 

The abbreviations of antimicrobials were updated according to the latest CLSI recommendations (CLSI M100-ED32:2022 Performance Standards for Antimicrobial Susceptibility Testing, 32nd Edition, http://em100.edaptivedocs.net).

7. L256 - The fact that strains cluster together cannot be used to determine their serotype, please amend the sentence accordingly.

Response:

As requested, the sentence was modified.

8. Paragraph L274 - 284 - Again please move this part to MM section, these are not results. 

Figure 4 - this figure is not needed and it is confusing. Please remove or move to supplementary information. This information is already found in the figure 3 and doesn't need to be repeated.

Response:

The background information and justification for WGS are given within the experimental context that also includes ERIC data. This part will not make any sense if it is isolated and moved elsewhere. Figures 3 and 4 report different sets of data, the former is produced with ERIC-PCR data and the latter - with WGS data.

9. Considering high number of AMR genes detected, please add supplementary table with the list of the genes, and the percentage identity and coverage. These unusual results have to be made available for revision and for the readers.

Response: The detailed AMR data have been presented by us earlier, please refer to our publication:

Sedrakyan, A.M.; Ktsoyan, Z.A.; Arakelova, K.A.; Zakharyan, M.K.; Hovhannisyan, A.I.; Gevorgyan, Z.U.; Mnatsakanyan, A.A.; Kakabadze, E.G.; Makalatia, K.B.; Chanishvili, N.A.; Pirnay, J.-P.; Arakelyan, A.A.; Aminov, R.I. Extended-Spectrum β-Lactamases in Human Isolates of Multidrug-Resistant Non-typhoidal Salmonella enterica. Front. Microbiol. 2020, 11, 592223. doi: 10.3389/fmicb.2020.592223.

All these questions are addressed in the above publication in great detail. This is an open access publication and thus can be accessed by anyone interested in the topic, even without access to academic libraries.

Columns with AMR data in Table S1 were removed from the current manuscript since they are not relevant to its scope.

10. L515-L519 - The authors did not analyse the animal reservoirs but only human strains, how can this be considered a weakness of the study then?

Response: This point that is made in Discussion refers to the difficulty of revealing animal reservoirs, not to the weakness of this study. This study is concerned with the human salmonellosis cases. 

11.L 559- 561 - serotype determination is not needed to antimicrobial resistance testing. It is sufficient to identify the strain to the species level to treat the patient according to the phenotypic susceptibility tests. Please remove the sentence.

Response:

It is clearly demonstrated in Fig. 3 that different NTS serotypes have different AMR/MDR profiles. For example, S. Typhimurium isolates are resistant to a broader range of antimicrobials, which also includes more frequent MDR phenotypes and the production of ESBL, compared to S. Enteritidis. If, for instance, a standard therapy with ceftriaxone is initiated, it may fail in the case of infection  by S. Typhimurium. This is why it is important to know the serotype. Besides, the serotype information can be obtained much faster than antibiograms. 

12. Paragraph L673 - please add parameters used for Prokka and the QC criteria adapted for quast and kraken analysis. Which thresholds were adopted to remove poor quality strains?

Response:

Analyses were performed with default settings and this is incorporated in the text (highlighted in yellow). Regarding the quality of the assembled genomes: this parameter was assessed using Genomics tools of the Bacterial and Viral Bioinformatics Resource Center (BV-BRC, https://www.bv-brc.org). The genome quality for all 42 isolates in this study was assessed as “Good” and already reported in the original submission (see Table S1). 

13. SeqSero has since a few years been superseded by \SeqSero2 which is more accurate and updated version, why did the authors use the old version? Please add whether reads or assemblies were used for the analysis.

Response: In the revision, we actually used SeqSero2, the text now is corrected. The use of short reads is of course inappropriate and we used assemblies as other researchers do.

14. L701 What was Blast used for, which blast? Blast nucleotide? Which parameters were used? Please add this information.

Response:

BLAST is  the Basic Local Alignment Search Tool, which is available online (https://blast.ncbi.nlm.nih.gov/Blast.cgi). It is used for nucleotide/protein similarity search. Since this database contains many gene and genome sequences of Salmonella, researchers in the field use blastn with default parameters. 

15. considering the high number of pargraphs, please add conclusion section adding the most important conclusions of the study. It is not sufficient that it can be found in the abstract.

Response: 

As requested, the most important conclusions of the study were added at the end of Discussion.
